# Three-Dimensional Spinal Evaluation Using Rasterstereography in Patients with Adolescent Idiopathic Scoliosis: Is It Closer to Three-Dimensional or Two-Dimensional Radiography?

**DOI:** 10.3390/diagnostics13142431

**Published:** 2023-07-20

**Authors:** Anne Tabard-Fougère, Charlotte de Bodman, Amira Dhouib, Alice Bonnefoy-Mazure, Stéphane Armand, Romain Dayer

**Affiliations:** 1Division of Pediatric Orthopaedics, Geneva University Hospitals, University of Geneva, 1205 Geneva, Switzerland; 2Department of Radiology, Reseau Hospitalier Neuchatelois, 2000 Neuchatel, Switzerland; 3Kinesiology Laboratory, Geneva University Hospitals and University of Geneva, 1205 Geneva, Switzerland

**Keywords:** adolescent idiopathic scoliosis, rasterstereography, biplanar radiography, 3D radiography, comparative study, scoliosis severity

## Abstract

(1) Background: Adolescent Idiopathic Scoliosis (AIS) is a three-dimensional (3D) spine deformity. The Cobb angle, evaluated with 2D radiography, is the gold standard to determine curve severity. The primary aim of this study was to evaluate the 3D spinal evaluation with rasterstereography in patients with AIS. The hypothesis was that rasterstereography reached higher accuracy than the gold standard 2D radiography. The second aim was to compare rasterstereography with 3D radiography. The hypothesis was that the rasterstereographic evaluation of patients with severe major scoliosis curves is closer to 3D radiography compared to the gold standard (2D radiography). (2) Methods: This is a prospective comparative study of a consecutive series of 53 patients, with the scoliosis curve evaluated with two 3D methods and the gold standard (2D radiography). (3) Results: The hypothesis that rasterstereography reached higher accuracy than the gold standard 2D radiography was validated for all curves. Even if all curves were highly correlated, both rasterstereography and 2D radiography scoliosis evaluation were underestimated for moderate/severe curves compared to 3D radiography. (4) Conclusions: The rasterstereographic evaluation of major curve scoliosis is not accurate enough to replace 2D radiography for moderate/severe curves. A longitudinal follow-up should be assessed in future studies to define the sensitivity of the detection of a significant change in the scoliotic mild and moderate curve (<40°).

## 1. Introduction

Adolescent Idiopathic Scoliosis (AIS) is a three-dimensional (3D) spine deformity that consists of the lateral deviation and axial rotation of the spine [1]. Even though it is a 3D spinal deformity, the gold standard for AIS surveillance is the Cobb angle evaluated with two-dimensional posterior-anterior full-length spine radiographs (2D radiography) [2,3]. This Cobb angle is the standard parameter used to quantify scoliosis curve magnitude [4]. However, the Cobb method that is used for decisions to observe, brace, or recommend surgical intervention for scoliosis has several limitations.

The first limitation is that the monitoring of spinal deformity evolution increases the frequencies of radiological assessments, which have negative long-term effects on young patients [5,6,7]. Indeed, the repeated use of radiography in the scoliosis monitoring context has been associated with a long-term increase in mortality from breast cancer [8]. This risk is potentially more important in children due to their greater sensitivity to irradiations [9]. Thus, the reduction of these irradiations is a major health challenge for these growing patients.

The second limitation is that because spinal deformity is a 3D deformity, not a 2D deformity, the Cobb angle is subject to errors [10]. The importance of the 3D evaluation of severe AIS has been reported in the literature. Lechner et al. [11] reported that the measured Cobb angle was on average around 10 degrees lower in the group evaluated with 2D radiography compared to those evaluated with a 3D CT scan. However, a 3D CT scan requires high radiation exposure, with 9.9 mSv compared to 0.1 mSv for chest radiography [12].

In this context, a measurement allowing us to reduce the radiation dose and reconstruct the 3D spinal structure is important in improving scoliosis evaluation.

In recent years, the EOS^®^ System (Biospace Med, Paris, France) developed a radiographic system which provides reduced irradiation (by 8–10 compared to standard radiographs) [13,14] and reconstructs the bone structure in 3D, allowing 3D spine reconstruction. The EOS^®^ System was validated in the clinical setting to follow the evolution of AIS patients [15,16,17]. A recent case series reported that the total radiation exposure was, however, only moderately reduced (50.6%) to skeletally immature scoliosis patients [18]. A recent study reported that only 60% of the axial spinal deformity can be determined by the frontal deformity in 200 AIS patients evaluated with biplanar radiography and respective 3D reconstruction [19]. Recently, Machida et al. [20] compared the 3D spine evaluation of coronal deformity for patients with severe AIS to 2D images. They reported proportional bias in the thoracic kyphosis measurement and lumbar lordosis, which refers to the underestimated values of these sagittal angles in the 2D method compared to 3D reconstruction.

At the same time, radiation-free methods have been increasingly used, such as raster-stereography, developed by Drerup and Hierholzer in the 1980s [21,22]. These methods measure 3D spine deformity using back surface topography. It consists of measuring the back shape of a patient in the standing position by projecting parallel lights onto the back. Distortions of the raster lines are then detected by a digital camera. After that, mathematical shape analysis enables the 3D reconstruction model of the spine [23,24,25] and the calculation of spinal parameters defining the frontal, sagittal, and frontal profiles. Among several commercial systems, the Formetric-4D system (Diers International GmbH, Schlangenbad, Germany) has been validated against 2D radiography by a collaborative study between the Diers company and German universities [23]. This system is largely used in research to measure and control spine deformities in AIS patients [26,27,28,29,30,31]. Most of the previous studies evaluating the validity of this system [31] concluded a good to strong correlation (r > 0.60) compared with 2D radiographs. To the best of our knowledge, its accuracy compared to 3D radiography has not been studied.

The primary aim of this study was to evaluate the 3D spinal evaluation with rasterstereography in patients with AIS. The hypothesis was that the rasterstereographic evaluation of patients with scoliosis curves reached a higher accuracy than the gold standard (2D radiography). The second hypothesis was that there is no proportional bias of major curve severity on this accuracy. The second aim was to compare rasterstereography with 3D radiography, assuming that 3D radiography is better than 2D radiography. The hypothesis was that the rasterstereographic evaluation of patients with severe major scoliosis curves is closer to 3D radiography than the gold standard (2D radiography).

## 2. Materials and Methods

### 2.1. Study Design

This is a prospective comparative study of a consecutive series of 53 patients, with scoliosis curves evaluated with two 3D methods and the gold standard (2D radiography).

### 2.2. Participants

Adolescents with AIS were recruited among patients programmed for radiological assessment with the EOS^®^ System for the suspicion or monitoring of scoliosis between 2014 and 2017. The local ethics committee approved this study (CER no. 13-255) and informed consent was obtained from all participants and their respective legal guardians.

The inclusion criteria were:Age between 10 and 18 years;AIS with Cobb Angle >10°, as measured with 2D radiography;Being programmed for radiological assessment with the EOS^®^ System for scoliosis monitoring.

The exclusion criteria were:History of spinal or thoracic surgery;Other neurological or orthopedic pathology;Being unable to stand upright;Tattoos or scarring on the back;Body Mass Index (BMI) ≥29.

### 2.3. Data Assessment

Each recruited participant had two evaluations in the same order during the same day: biplanar 2D radiography and rasterstereographic evaluation.

The participant underwent postero-anterior biplanar radiography performed with the EOS^®^ System as part of his clinical visit. The participant was in a standing position, allowing the spine to be examined under normal weight-bearing conditions. The feet were on the same alignment in the frontal plane, with 20–25 cm between the two feet. To decrease the artefacts due to the projection of the humerus on the spine in the lateral view, without modifying the spine shape, the fists were placed on the clavicles [32]. Then, the participant’s specific 3D spinal reconstructions were obtained using the dedicated validated software SterEOS^®^ (SterEOS, Biospace Med, Paris, France), operated by a trained physician to produce 3D reconstructions of the spine and rib cage [33].

Immediately after radiography evaluation, the rasterstereographic Formetric-4D System with the DICAM software (version 2, Diers International GmbH, Schlangenbad, Germany) was used to estimate spinal angles. The patient’s upper body was undressed, with the pants pulled down under the buttocks, and the patient’s hair was tied up to the hairline to make the neck visible. Reflective markers were manually placed at the level of the vertebra prominent, right and left lumbar dimples, and apical vertebra for patients with obvious scoliosis. Measurements were made in the standing position with the fists on clavicles. The modality used was 3D-static, which allows measurements based on one image during a one-second capture.

### 2.4. Data Analysis

The primary outcome was the Cobb angle (°), as provided by 3D radiography (EOS^®^ System) and its equivalent measured with rasterstereography (Formetric-4D System), called the scoliosis angle (°) (Figure 1). Concerning 3D radiography, the Cobb angle was automatically measured, as described by Gille et al. [34]. Concerning rasterstereography, the scoliosis angle was automatically measured between tangents to the cranial and caudal endplates of the calculated cranial and caudal vertebral bodies, respectively [35]. The secondary outcomes were thoracic kyphosis from T4 to T12 and lumbar lordosis from L1 to L5.

These outcomes were extracted automatically with rasterstereography and 3D radiography. Finally, the same outcomes were blindly measured by one pediatric spine surgeon on a full-standing posteroanterior 2D radiography (gold standard) in clinical routine.

### 2.5. Statistical Analysis

First, the scoliosis angle and Cobb angle, as measured, respectively, with rasterstereography and 3D and 2D radiography, were compared using paired Student *t*-tests. In addition, the mean differences between the measurements provided by each system were evaluated and the number of participants with a difference value superior to 5 degrees was counted between each system; the 5° margin of error of the Cobb angle measurement was performed on radiography [36,37,38]. The correlations between the three methods of measurement were computed with the Pearson correlation coefficient (r), with respective 95% confidence intervals (95% CI).

The agreement and reliability between the three methods were assessed using Bland–Altman plots and the single measure two-way intraclass correlation coefficient (ICC) with equation [39]. The presence of proportional bias was reported by assessing the slope regression line fitted to the Bland–Altman plot.

Analyses were performed using R software (version 4.2.2, R Development Core Team, 2018) and the RStudio interface (RstudioTeam (2015), Rstudio: Integrated Development for R.Rstudio, Inc., Boston, MA, USA; http://www.rstudio.com/ accessed on 8 May 2023). The interpretation criteria for an agreement were defined as follows: an ICC > 0.90 was described as excellent, an ICC between 0.75 and 0.90 was good, an ICC between 0.50 and 0.57 was moderate, and an ICC < 0.50 was poor [40]. The interpretation criteria for correlation were defined as follows: an r > 0.80 was strong, an r between 0.60 and 0.80 was moderate, an r between 0.30 and 0.60 was fair, and an r < 0.30 was poor [41].

## 3. Results

### 3.1. Population Description

Fifty-three adolescents were included with 32 females between 2014 and 2017. The mean age was 13.5 ± 1.9 years. The mean BMI was 19.1 ± 3.4 kg/m^2^. The average Cobb angle was 23.9 ± 16.4°, as evaluated with 2D radiography (gold standard), including 33 (62%) patients with mild curve severity (Cobb < 25°) and 20 (38%) with moderate (n = 9) and severe (n = 11) curves. A total of two curves were localised in the lumbar region (4%), 32 (60%) curves in the thoraco-lumbar region, and 19 (36%) curves in the thoracic region (Table 1).

### 3.2. Comparison between Rasterstereography and 2D Radiography

There was no significant difference (*p* = 0.127) in the mean measurement of the scoliosis angle evaluated with rasterstereography (25.7 ± 17.0°) and 2D radiography (24.1 ± 16.1°), and the mean difference between them was 6.4 ± 4.7° (n = 25 (47%) patients with difference >5°). The correlation was strong (r = 0.89) and the agreement was excellent (ICC = 0.94) (Table 2, column 1, and Figure 2A).

Similarly, there was no significantly different scoliosis angle evaluated with rasterstereography and 2D radiography in the mild curves (*p* = 0.056), with moderate correlation (r = 0.63) and moderate agreement (ICC = 0.75). In addition, no significant difference in the scoliosis angle between rasterstereography and 2D radiography in moderate/severe curves was reported (*p* = 0.624), with moderate correlation (r = 0.64) and good agreement (ICC = 0.77).

As reported in Figure 2D, the presence of proportional bias was not graphically highlighted and the agreement limits were between 17° and −13°, with a 1.6° mean value. A total of five patients (56%) with moderate curves (25–40 degrees) had an overestimated scoliosis angle (>10 degrees) with rasterstereography compared to 2D radiography.

### 3.3. Comparison between Rasterstereography and 3D Radiography

There was no significant difference in the mean measurement of the scoliosis angle evaluated (rasterstereography: 25.7 ± 17.0°, 3D radiography: 27.2 ± 23.7°, *p* = 0.227), and the mean difference was 7.5 ± 6.4° (n = 29 (55%) patients with difference > 5°). The correlation was strong (r = 0.93) and the agreement was excellent (ICC = 0.94) (Table 2, column 2, and Figure 2B).

There was a significantly higher scoliosis angle evaluated with rasterstereography (14.6 ± 6.1°) compared to 3D radiography (10.6 ± 7.0°) in the mild curves (*p* < 0.001), with moderate correlation (r = 0.77) and good agreement (ICC = 0.79). On the other hand, a significantly lower scoliosis angle was evaluated with rasterstereography (43.0 ± 14.2°) compared to 3D radiography (53.8 ± 15.0°) in moderate/severe curves (*p* < 0.01), with strong correlation (r = 0.83) and good agreement (ICC = 0.76).

As reported in Figure 2E, the presence of proportional bias was graphically highlighted with a scoliosis angle computed with 3D radiography underestimated for mild curves and overestimated for moderate/severe curves compared to rasterstereography. The agreement limits were between 21°and −17.6°, with a −3.3° mean value.

### 3.4. Comparison between 3D and 2D Radiography

There was a significantly higher scoliosis angle mean value (*p* = 0.026) evaluated with 3D radiography (27.2 ± 23.7°) compared to 2D radiography (23.9 ± 16.4°), and the mean difference was 7.8 ± 7.6° (n = 24 (45%) patients with difference > 5°). The correlation was strong (r = 0.93) and the agreement was excellent (ICC = 0.93) (Table 2, column 3).

There was a significantly lower scoliosis angle evaluated with 3D radiography (10.6 ± 7.0°) compared to 2D radiography (12.6 ± 6.1°) in the mild curves (*p* = 0.008), with strong correlation (r = 0.82) and good agreement (ICC = 0.88). Conversely, a significantly higher scoliosis angle was evaluated with 3D radiography (53.8 ± 15.0°) compared to 2D radiography (41.9 ± 10.2°) in moderate/severe curves (*p* < 0.001), with moderate correlation (r = 0.61) and poor agreement (ICC = 0.47).

As reported in Figure 2F, the presence of proportional bias was graphically highlighted with a scoliosis angle computed with 3D radiography underestimated for mild curves and overestimated for moderate/severe curves compared to 2D radiography. The agreement limits were between 23.6° and −17.2°, with a −1.7° mean value.

## 4. Discussion

This is the first prospective comparative study of a consecutive series of AIS patients with scoliosis curves evaluated with two 3D methods and the gold standard that is 2D radiography.


*Primary objective*


The primary aim of this study was to evaluate the three-dimensional spinal evaluation with rasterstereography in patients with AIS. The hypothesis that rasterstereography reached a high accuracy compared to the gold standard 2D radiography was validated for all curves.

Indeed, the present results showed that scoliosis and Cobb angles, as evaluated by rasterstereography, were strongly correlated (r > 0.80) with 2D radiography, with no significant difference (6.4° mean difference). These values are near the 5° margin of error of Cobb angle measurement performed on radiography [36,37,38], with almost 50% of curves with more than 5° of difference between rasterstereography and 2D radiography. These results confirmed what the existing literature reported in a recent systematic review [31], with a good to strong correlation (r > 0.60) between rasterstereography and 2D radiography. The mean difference values are also in line with the reported mean difference in the existing literature, ranging from 5.4° to 8.8° [27,35].

The second hypothesis that there is no proportional bias of major curve severity on this accuracy was validated. As reported in Figure 2D, the presence of proportional bias was not graphically highlighted and the agreement limits were between 17° and −13°, with a 1.6° mean value.


*Secondary objective*


The hypothesis that rasterstereography is closer to 3D radiography compared to the gold standard (2D radiography) was not validated considering the proportional bias due to curve severity. Both rasterstereography and 2D radiography scoliosis evaluation were underestimated for moderate/severe curves and overestimated for mild curves compared to 3D radiography. This proportional bias was not reported when comparing rasterstereography and 2D radiography. On the contrary, 3D radiography evaluation differed more from the gold standard than rasterstereography.

The present results are in line with the results of Lechner et al. [11], who reported a Cobb angle that was on average 10 degrees larger in the 3D group (evaluated with 3D CT scan) than in the 2D radiography group. The present results are also in line with the results of Machida et al. [20], who reported proportional bias in a thoracic kyphosis measurement and lumbar lordosis, which refers to the underestimated values of these sagittal angles in a 2D method compared to the 3D reconstruction. Even if the study design did not make it possible to evaluate if the 3D radiography is closer to the real spinal curvature than the 2D gold standard, it is intuitive to accept that a 2D outcome could not accurately represent a 3D deformity, especially for severe curves.

In the present study, the operator who performed the 3D reconstruction reported the difficulty of the identification and lineation of pedicle and spinous processes for patients with severe curves, due to visual vertebral superposition. In this context, the cobb angle of severe curves evaluated with 3D radiography could be erroneous. The inter-operator reliability of the 3D radiography reconstruction was reported as good in the current literature, with the reliability of the axial rotation in the upper and middle thoracic spine worth noting [42]. This was attributed to the obtuse angulation of pedicle and spinous processes in the frontal view, which make their identification and delineation difficult. However, the measurement error of 3D radiography compared to the 3D CT scan, evaluated with a phantom scoliotic curvature, showed good accuracy with only 1.6° (max = 3.9°) deviations of the Cobb angle [43]. The phantom scoliotic curve did not replicate real clinical conditions. These results should be confirmed with further studies through a comparison of real patients and severe scoliosis major curves evaluated with a 3D CT scan.


*Perspective for therapeutic attitude*


Both rasterstereography and 3D radiography could be used for the AIS monitoring of curves inferior to 25°, as 2D radiography is currently used for it. However, AIS cannot be fully evaluated without 2D radiography. During the first consultation, radiography must be maintained to ensure the integrity of vertebral morphology, and exclude, for example, a congenital scoliosis. The purpose of developing a radiation-free mode of scoliosis surveillance is primarily to reduce the number of required radiographs during childhood. However, the accuracy for monitoring curve progression should be evaluated with a follow-up design.

Indeed, the repeated use of radiation, especially in children, because of their greater sensitivity to radiation, has been shown to be deleterious to their future health [7,44]. For the mild curves (<25°), several attempts have been made to validate non-invasive measurement tools. For example, the inclinometer [45] or the scoliometer (National Scoliosis Foundation, Watertown, MA, USA) [46] were proposed as a radiation-free alternative to plain radiographs. Recently, the early diagnosis of AIS with rasterstereography showed better diagnostic characteristics compared to the angle of trunk rotation evaluated with a scoliometer [47].

Although no patient can avoid radiographs completely for the moment, there should be an effort to reduce exposure whenever possible. The benefits of this evaluation method in reducing x-ray exposure in children and adolescents are readily apparent. Radiation-free techniques for monitoring scoliosis changes should be actively developed and validated.

There are several advantages to rasterstereography. It is easily and quickly done. It is less expensive than low-dose radiography (2D or 3D) and less time consuming. The time for reconstruction for 3D radiography is about 15 to 30 min [14], whereas the time for acquisition with rasterstereography is less than 5 min with only one operator. This suggests that rasterstereography could be a better first-line procedure in the assessment of surveillance for curves less than 25°, as evaluated by 2D radiography. However, for severe curves, 3D radiography should be used to accurately evaluate the major curve scoliosis angle. It seems that the rasterstereographic evaluation of major scoliosis curves is not accurate enough to replace 2D radiography for moderate/severe curves. However, according to the difficulty reported in the present study with 3D radiography reconsctruction (identification and lineation of pedicle and spinous processes) for patients with severe curves, these results should be confirmed with a 3D CT scan comparison.

Finally, the 3D spine evaluation provides supplementary information that complements the Cobb angle. Rasterstereography and 3D radiography can provide many additional parameters with no additional time and independent of the operator (back surface rotation, waist crease vertical asymnetry, rib prominence column, etc.). The patient’s self-perception should also be considered in addition to the Cobb angle. Thakur et al. recently reported that several 3D surface topographic outcomes (shoulder asymmetry, scapula/lumbar/pelvic regions) were highly correlated to patient self-image questionnaires [48].


*Limitations and perspectives*


The first limitation of this study is that the number of patients with moderate/severe curves who were assessed is relatively small. The moderate curves (25–40°) are particularly not represented enough in the present study, with only 9 patients. A total of five patients (56%) had overestimated the scoliosis angle (>10°) with rasterstereography compared to 2D radiography. These patients are typically treated with a brace. Thus, the impact of the brace on surface topography evaluation should be evaluated in further studies with a larger sample size of moderate curves. The second limitation is due to the difficulty of 3D radiography reconsctruction in patients with severe curves; these results should be confirmed by comparison with 3D CT scans.

The next step is to investigate the accuracy of rasterstereography in detecting curve progression with a longitudinal follow-up study in both patients with Cobb angles less than 25° and between 25° and 40°. The purpose of surveillance is primarily to identify a change. This change monitored with rasterstereography will alert the clinician of the possibility of progression in the true scoliosis curve for patients with a Cobb angle inferior to 25°. Because the consequences of misdiagnosing a scoliosis of 25° or more can be devastating for the future health of a skeletally immature patient, a radiation-free test must be highly sensitive.

## 5. Conclusions

In conclusion, the hypothesis that rasterstereography reached a higher accuracy than the gold standard 2D radiography was validated for all curves. Even if all curves were highly correlated, both rasterstereography and 2D radiography scoliosis evaluations were underestimated for moderate/severe curves compared to 3D radiography. Both rasterstereography and 3D radiography could be used for the AIS surveillance of curves inferior to 25°, as 2D radiography is currently used for it. The rasterstereographic evaluation of major curve scoliosis is not accurate enough to replace 2D radiography for moderate/severe curves. A longitudinal follow-up should be assessed in future studies to define the sensitivity of the detection of a significant change in the scoliotic mild and moderate curve (<40°).

## Figures and Tables

**Figure 1 diagnostics-13-02431-f001:**
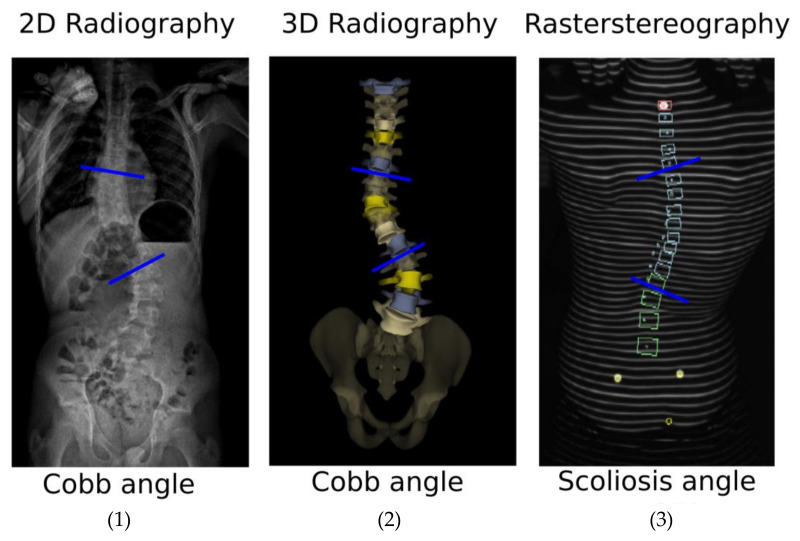
Illustration of the primary outcome (major scoliosis curve angle) evaluated on the same patient with respectively: (1) 2D radiography, (2) 3D radiography, and (3) 3D rasterstereography. The blue lines correspond to the upper and lower end vertebra endplate used to evaluate the primary outcome.

**Figure 2 diagnostics-13-02431-f002:**
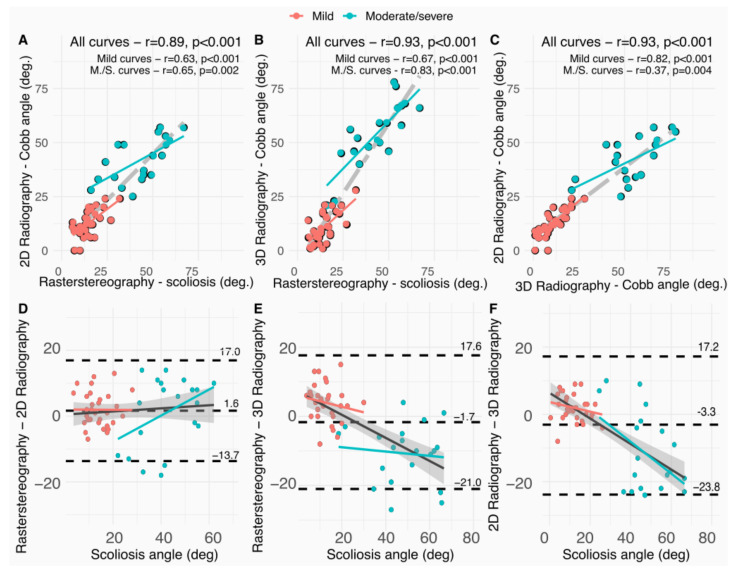
The first line illustrated the correlation (r is Pearson correlation coefficient) of major scoliosis curves evaluated with each system. The second line illustrated the Bland–Altman plots with the slope regression line fitted to the Bland–Altman plot: (**A**,**D**) rasterstereography vs. 2D radiography, (**B**,**E**) rasterstereography vs. 3D radiography, and (**C**,**F**) 3D vs. 2D radiography. The grey area correspond to the 95% confidence intervals of the slope regression line fitted to the Bland-Altman plot. The dashed lines correspond to the mean difference and the 95% agreement limits (±1.96 SD).

**Table 1 diagnostics-13-02431-t001:** Description of the population.

	All Patients (n = 53)	Mild Curve (n = 33)	Moderate/Severe Curve (n = 20)
Age, years	13.5 (1.9)	13.2 (1.8)	14.2 (1.9)
Female, n (%)	33 (62%)	19 (58%)	14 (70%)
BMI, kg/m^2^	19.1 (3.4)	18.5 (3.0)	20.0 (3.9)
Height, m	1.62 (0.10)	1.63 (0.09)	1.62 (10.9)
Weight, kg	50.9 (11.1)	49.6 (10.6)	53.1 (11.8)
Cobb angle (2D), deg.	23.9 (16.4)	12.6 (6.1)	41.9 (10.2)
Curve location			
Thoracic, n (%)	19 (36%)	15 (45%)	4 (20%)
Thoraco-lumbar, n (%)	32 (60%)	16 (48%)	16 (80%)
Lumbar, n (%)	2 (4%)	2 (3%)	0 (0%)
Thoracic kyphosis, deg.	33.7 (10.8)	35.0 (10.0)	31.2 (12.0)
Lumbar lordosis, deg.	43.3 (11.3)	41.7 (10.4)	46.1 (12.6)

Mild curve is defined with a Cobb angle inferior to 25 degrees manually evaluated on 2D XR, and moderate/severe curve is superior or equal to 25 degrees. BMI is body mass index, m is meter, kg is kilograms, n is the number of participants, deg. is degrees. Thoracic kyphosis angle is manually evaluated on 2D radiography from T4 to T12. Lumbar lordosis angle is manually evaluated on 2D radiography from L1 to L5.

**Table 2 diagnostics-13-02431-t002:** Systems’ comparison results of major curve scoliosis angle for all, mild, and moderate/severe (M./S.) curves.

		Raster vs. 2D XR	Raster vs. 3D XR	3D vs. 2D XR	
Major curve scoliosis angle					
Mean diff. deg.	All (53)	6.4 (4.7)	7.5 (6.4)	7.8 (7.6) *	
	Mild (33)	4.3 (3.5)	5.2 (3.9) **	3.5 (2.7) **	
	M./S. (20)	9.7 (4.7)	11.2 (7.9) **	14.8 (7.9) **	
Diff. > 5 deg n (%)	All (53)	25 (47%)	29 (55%)	24 (45%)	
	Mild (33)	10 (30%)	15 (45%)	8 (24%)	
	M./S. (20)	15 (75%)	14 (70%)	16 (80%)	
Diff. > 10 deg n (%)	All (53)	11 (21%)	12 (23%)	12 (23%)	
	Mild (33)	2 (6%)	4 (13%)	0 (0%)	
	M./S. (20)	9 (45%)	8 (40%)	10 (40%)	
Correlation r (95% CI)	All (53)	0.89 (0.81; 0.94)	0.93 (0.89; 0.96)	0.93 (0.88; 0.96)	
	Mild (33)	0.63 (0.36; 0.80)	0.77 (0.57; 0.87)	0.82 (0.66; 0.91)	
	M./S. (20)	0.64 (0.28; 0.84)	0.83 (0.61; 0.93)	0.61 (0.23; 0.83)	
Agreement ICC (95% CI)	All (53)	0.94 (0.90; 0.97)	0.94 (0.90; 0.97)	0.92 (0.87; 0.96)	
	Mild (33)	0.75 (0.49; 0.88)	0.79 (0.59; 0.90)	0.88 (0.75; 0.94)	
	M./S. (20)	0.77 (0.42; 0.91)	0.76 (0.41; 0.91)	0.47 (−0.33; 0.79)	

Values were compared using paired Student *t*-tests. Level of significance was reported as * for *p*-value < 0.05 and ** for *p*-value < 0.01. Deg. is degrees, ICC is the single measure two-way intraclass correlation coefficient, r is the Pearson correlation coefficient, 95% CI is 95% confidence intervals, mean diff is the mean difference between each system, XR is radiography, and raster is rasterstereography.

## Data Availability

Not applicable.

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
