# Peer review of "Three-Dimensional Spinal Evaluation Using Rasterstereography in Patients with Adolescent Idiopathic Scoliosis: Is It Closer to Three-Dimensional or Two-Dimensional Radiography?"

_diagnostics, 2023, doi:10.3390/diagnostics13142431_

Round 1

Reviewer 1 Report

Title: Three-Dimensional spinal evaluation using rasterstereography in patients with adolescent idiopathic scoliosis: is it closer to 3-Dimensional than 2-Dimensional radiography?

Outline: This study is aimed to confirm that rasterstereography is closer to 3D radiography compared to the gold standard 2D radiography. From this paper, the authors concluded that both rasterstereography and 3D radiography, could be used for AIS surveillance of mild curves, as 2D radiography is currently used for it. The rasterstereographic evaluation of major curve scoliosis is not enough accurate to replace the 2D radiography for moderate/severe curves.

Critique

As one of the methods to diagnose as AIS, it can be one of the options for diagnostic tool because adolescent anywhere can have potentially exposure the radiological hazard. Thus, it is new options so has archival values.

However, I have one thing to question about the paper for the merits. All of AIS patients need to capture 2D radiography. So, in view of the effectiveness, Do the authors think that AIS can be fully evaluated without 2D x-ray testing?

Author Response

We would like to thank the reviewer for the comments and for the appreciation of the importance of our work.

I do not think AIS can be fully evaluated without 2D x-ray. During the first consultation, radiography must be performed to ensure integrity of the vertebral morphology and exclude for example a congenital scoliosis. Then both rasterstereography and 3D radiography could be used for AIS monitoring of curves inferior to 25°, as 2D radiography is currently used for it. However, the accuracy for monitoring curve progression should be evaluated with a follow-up design.

Concerning AIS screening, the use of rasterstereography for early detection of AIS patients has shown better results than scoliometer (Vendeuvre et al. 2023). However, these results recently published only included children and adolescents referred for a suspected AIS. Future screening studies should validate these results.

Reviewer 2 Report

In my opinion, the main issue concerns the comparison of a gold standard (2D X-ray) with two "other" methods. Rasterstereography demonstrated moderate accuracy in measuring the scoliosis degree and low accuracy in monitoring the curve progression. Accordingly, it cannot be considered as a valid alternative to radiographic evaluavtion. However, since demonstrated capable of revealing the presence of spine deformity, it could be in principle considered for the early screening in large adolescent populations (Bassani et al.). On the other side, to the best of my knowledge, 3D EOS has never been compared with CT-Scan for measuring axial rotation. It has been suggested that the "real" curve is more precisely measured with 3D EOS than 2D X-Ray. However, since 3D EOS is a reconstruction, it is difficult to confirm this insight without comparison with a real gold standard, e.g. 3D CT Scan, which is not possible in clinical settings due to high radiation.

Thus, the main hypothesis (rasterstereographic evaluation of patient with moderate/severe major scoliosis curves are closer to 3D radiography compared to the gold standard) seems inadapted.

However, the data are interesting and the analysis is well performed.

Therefore, I suggest the main question of this article should be rebuilt according to these concepts. For example, you should write that the main hypothesis was that rasterstereographic evaluation of patient with scoliosis curves reached high accuracy, as compared with the gold standard, and the second hypothesis was that the accuracy was better for severe curves of for mild curves, as you wish. The secondary aim should be to compare rasterstereography with 3D EOS, with a 3rd hypothesis assuming one is better. 

Not too bad but I suggest English editing. 

Author Response

We would like to thank the reviewer for the comments and for the appreciation of the importance of our work.

As suggested, the main research question was reformulated as follows:

“The primary aim of this study was to evaluate the 3D spinal deformity with ras-terstereography in patients with AIS. The hypothesis was that rasterstereographic evaluation of patients with scoliosis curves reached high accuracy, as compared to the gold standard (2D radiography). The second hypothesis was that there is no proportional bias of major curve severity on this accuracy. The second aim was to compare rasterstereography with 3D radiography, assuming that 3D radiography is better than 2D radiography. The hypothesis was that rasterstereographic evaluation of patient with severe major scoliosis curves are closer to 3D radiography compared to the gold standard (2D radiography).”

Comments on the Quality of English Language: Not too bad but I suggest English editing.

Given the fact that the comments of the other reviewer were that English language was fine with no issues detected, we kindly ask the Editor if he want us to provide professional English editing to our work.

Round 2

Reviewer 2 Report

I suggest minor, formal revisions.

I would write differently "The first one is because..." and "The second limitation is because". 

The conclusion should also be rewritten according to the new research questions.

I would write differently "The first one is because..." and "The second limitation is because". 

Author Response

Reviewer #2

Comments and Suggestions for Authors

I suggest minor, formal revisions.

I would write differently "The first one is because..." and "The second limitation is because".

We would like to thank the reviewer for the comments and for the appreciation of the importance of our work.

As suggested, the limitation section was reformulated as follows:

“The first limitation of this study is that the number of patients with moderate/severe curves who were assessed is relatively small.  … The second limitation is due to the difficulty of 3D radiography reconstruction in patients with severe curves; these results should be confirmed by comparison with 3D CT scans.

The next step is to investigate the accuracy of rasterstereography in detecting curve progression with a longitudinal follow-up study in both patients with Cobb angles less than 25 and between 25 and 40. ...”

The conclusion should also be rewritten according to the new research questions.

As suggested, the conclusion was reformulated as follows:

“In conclusion, the hypothesis that rasterstereography reached high accuracy compared to the gold-standard 2D radiography was validated for all curves. Even if all curves were highly correlated, both rasterstereography and 2D radiography scoliosis evaluations were underestimated for moderate/severe curves compared to 3D radiography. ...”